Early archosauromorph remains from the Permo-Triassic Buena Vista Formation of north-eastern Uruguay

Ezcurra Martín D. 1 martindezcurra@yahoo.com.ar
Velozo Pablo 2
Meneghel Melitta 3
Piñeiro Graciela 2
1 School of Geography, Earth and Environmental Sciences, University of Birmingham , Edgbaston, Birmingham , UK
2 Departamento de Evolución de Cuencas, Facultad de Ciencias , Iguá, Montevideo , Uruguay
3 Laboratorio de Sistemática e Historia Natural de Vertebrados, Facultad de Ciencias , Iguá, Montevideo , Uruguay
Young Mark
Electronic publication date: 2015 Feb 19
Publication date: 2015
Volume: 3
Electronic Location ID: e776
Received 2014 Dec 9; Accepted 2015 Jan 28
Copyright: © 2015 Ezcurra et al.
Copyright year: 2015
Copyright holder: Ezcurra et al.
License: This is an open access article distributed under the terms of the Creative Commons Attribution License, which permits unrestricted use, distribution, reproduction and adaptation in any medium and for any purpose provided that it is properly attributed. For attribution, the original author(s), title, publication source (PeerJ) and either DOI or URL of the article must be cited.
License URL: https://creativecommons.org/licenses/by/4.0/

Keywords: Diapsida, Archosauromorpha, Permian, Proterosuchidae, Extinction, South America, Palaeobiogeography

Funding: Emmy Noether Programme Award from the Deutsche Forschungsgemeinschaft BU 2587/3-1 Marie Curie Career Integration Grant PCIG14-GA-2013-630123 The research of MDE was funded by Emmy Noether Programme Award from the Deutsche Forschungsgemeinschaft (BU 2587/3-1 to Richard Butler) and by a Marie Curie Career Integration Grant (PCIG14-GA-2013-630123 to Richard Butler). ANII_FCE2011_6450 supported the research of GP, MM and PV. The funders had no role in study design, data collection and analysis, decision to publish, or preparation of the manuscript.

==============================
The Permo-Triassic archosauromorph record is crucial to understand the impact of the Permo-Triassic mass extinction on the early evolution of the group and its subsequent dominance in Mesozoic terrestrial ecosystems. However, the Permo-Triassic archosauromorph record is still very poor in most continents and hampers the identification of global macroevolutionary patterns. Here we describe cranial and postcranial bones from the Permo-Triassic Buena Vista Formation of northeastern Uruguay that contribute to increase the meagre early archosauromorph record from South America. A basioccipital fused to both partial exoccipitals and three cervical vertebrae are assigned to Archosauromorpha based on apomorphies or a unique combination of characters. The archosauromorph remains of the Buena Vista Formation probably represent a multi-taxonomic assemblage composed of non-archosauriform archosauromorphs and a ‘proterosuchid-grade’ animal. This assemblage does not contribute in the discussion of a Late Permian or Early Triassic age for the Buena Vista Formation, but reinforces the broad palaeobiogeographic distribution of ‘proterosuchid grade’ diapsids in Permo-Triassic beds worldwide.

Introduction

Archosauromorpha is one of the major groups of diapsid reptiles, which includes around 10,000 species of living birds and crocodilians (Clements, 2007) and all extinct species more closely related to these extant groups than to lepidosaurs (Dilkes, 1998). The oldest known archosauromorphs are represented by four nominal species restricted to Upper Permian rocks of Europe and Africa (Protorosaurus speneri Von Meyer, 1832, Archosaurus rossicus Tatarinov, 1960, Eorasaurus olsoni Sennikov, 1997 and Aenigmastropheus parringtoni Ezcurra, Scheyer & Butler, 2014: Von Meyer, 1830; Sennikov, 1988; Gower & Sennikov, 2000; Gottmann-Quesada & Sander, 2009). In the aftermath of the Permo-Triassic mass extinction, the archosauromorph fossil record is considerably more abundant and morphologically diverse, including members of Rhynchosauria, Prolacertiformes, Proterosuchidae and Archosauria (Charig & Reig, 1970; Charig & Sues, 1976; Carroll, 1976; Dilkes, 1998; Gower & Sennikov, 2000; Butler et al., 2011; Nesbitt, 2011; Ezcurra, Butler & Gower, 2013). The Permo-Triassic (latest Permian-earliest Triassic) archosauromorph record is crucial to understand the impact of the Permo-Triassic mass extinction on the group and their subsequent taxonomic dominance in Mesozoic terrestrial ecosystems. However, the Permo-Triassic archosauromorph record is currently poor or essentially non-existent in several continents (South America, North America, Australia and Antarctica) (Camp & Banks, 1978; Thulborn, 1979; Thulborn, 1986; Colbert, 1987; Smith et al., 2011; Ezcurra, Butler & Gower, 2013; Ezcurra, Scheyer & Butler, 2014; Ezcurra, 2014). In particular, the South American Permo-Triassic archosauromorph record is remarkably scarce, being restricted to isolated postcranial bones from the Early Triassic Sanga do Cabral Formation of southern Brazil (Dias-da-Silva, 1998; Langer & Schultz, 1997; Langer & Lavina, 2000; Da-Rosa et al., 2009; Dias-da-Silva & Da-Rosa, 2011—the archosauriform Koilamasuchus gonzalezdiazi Ezcurra, Lecuona & Martinelli, 2010 from the Quebrada de los Fósiles Formation of central-western Argentina (Bonaparte, 1981) was recently redated as Middle-Late Triassic (Ottone et al., 2014)). Here, we increase the South American early archosauromorph record with the description of cranial and postcranial bones from the Permo-Triassic Buena Vista Formation of north-eastern Uruguay.

Geological and palaeontological setting

The Buena Vista Formation crops out in north-eastern Uruguay and consists of red-brownish sandstones, intercalated with thin layers of red-brownish mudstones and intraformational conglomerates deposited under continental fluvial conditions (Bossi & Navarro, 1991; Goso et al., 2001; Piñeiro & Ubilla, 2003). Most of the fossils collected from the Buena Vista Formation are found in the intraformational conglomerates close to the Colonia Orozco town and correspond to isolated bones to ocassionally articulated partial skeletons (Fig. 1). The tetrapod fossil content of the Buena Vista Formation represents the Colonia Orozco Local Fauna and includes laidleriid (Uruyiella liminea Piñeiro, Marsicano & Lorenzo, 2007), mastodonsaurid, rhinesuchid-like and dvinosaurian temnospondyls (Marsicano, Perea & Ubilla, 2000; Piñeiro, 2004; Piñeiro, Marsicano & Damiani, 2007; Piñeiro et al., 2007; Piñeiro, Ramos & Marsicano, 2012), procolophonoid parareptiles (i.e., Pintosaurus magnidentis Piñeiro, Rojas & Ubilla, 2004; Piñeiro, 2004), probable varanopid and sphenacodontid synapsids (Piñeiro & Ubilla, 2003; Piñeiro et al., 2013), and basal archosauromorphs (Piñeiro & Ubilla, 2003; Piñeiro, 2004) (present study). The age of the Buena Vista Formation is poorly constrained because of the absence of index taxa and the presence of taxa that are documented in either Late Permian or earliest Triassic assemblages. Therefore, the age of this formation has been substantially debated. The Buena Vista Formation has been considered a lateral equivalent of the Sanga do Cabral Formation of southern Brazil based on lithostratigraphic similarities (Andreis, Bossi & Montardo, 1980; Bossi & Navarro, 1991). The Brazilian unit is considered late Induan—early Olenekian in age because of the presence of the index taxon Procolophon (Dias-da-Silva, Modesto & Schultz, 2006) and, as a result, the same age has been assigned to the Buena Vista Formation (Bossi & Navarro, 1991). However, subsequent authors have suggested an older age for the Buena Vista Formation, being closer to the Permo-Triassic boundary or even within the Late Permian based on its tetrapod fossil content (Piñeiro & Ubilla, 2003; Piñeiro et al., 2003; Piñeiro, 2004; Piñeiro, Rojas & Ubilla, 2004; Piñeiro, Marsicano & Lorenzo, 2007; Piñeiro, Marsicano & Damiani, 2007; Piñeiro et al., 2007; Piñeiro, Ramos & Marsicano, 2012). In particular, the description of varanopid synapsids would favour a Permian age (Piñeiro et al., 2003), but recent authors have casted doubts on these assignments and concluded that there is no compelling evidence to support a Permian age over an Early Triassic one (Dias-da-Silva, Modesto & Schultz, 2006). Here, we will consider the Buena Vista Formation as Permo-Triassic in age, taking into account recent studies that placed the Colonia Orozco Local Fauna as a transitional assemblage that could contain the Permo-Triassic boundary (see Piñeiro, Ramos & Marsicano, 2012).

Figure 1 Maps showing the locality that yielded the archosauromorph remains described here (A) and stratigraphic column of the Buena Vista Formation as it outcrops in the locality indicating the archosauromorph-bearing levels with an asterisk (B).

Modified from Piñeiro, Ramos & Marsicano (2012).

Systematic Palaeontology

DIAPSIDA Osborn, 1903 sensu Laurin, 1991

SAURIA Gauthier, 1984 sensu Gauthier, Kluge & Rowe, 1988

ARCHOSAUROMORPHA Von Huene, 1946 sensu Dilkes, 1998

Gen. et sp. indet.

Figures 2, 3D–F, 4A, B, 5 and 6

Materials. FC-DPV 2641: co-ossified basioccipital and exoccipitals (Figs. 2, 3D–3F, 4A and 4B); FC-DPV 2640: anterior cervical vertebra (Figs. 5A–5C); FC-DPV 2639: middle or posterior cervical vertebra (Fig. 6); FC-DPV 2637: middle or posterior cervical vertebra (Figs. 5D–5H).

Horizon and Locality. Locality close to Colonia Orozco town, intraformational conglomerates of the Buena Vista Formation (Colonia Orozco Local Fauna, Permo-Triassic age, see geological and palaeontological setting), Cerro Largo County, north-eastern Uruguay (Fig. 1).

Figure 2 Partial braincase (FC-DPV 2641) from the Late Permian–Early Triassic Buena Vista Formation (Uruguay) in (A) posterior; (B) anterior; (C) right lateral; (D) left lateral; (E) dorsal; and (F) ventral views.

Abbreviations: bt, basal tubera; dr, diagonal ridge; ecf, endocranial floor; eo, exoccipital; fpbs, facet for the parabasisphenoid; fvro, facet for the ventral ramus of the opisthotic; lf, lateral flange of the basioccipital; mlr, median longitudinal ridge; mwpr, medial wall of the pseudolagenar recess; np, notochordal pit; oc, occipital condyle; rs, recessed surface; wmf, wall of the metotic foramen. Scale bar equals 5 mm.

Figure 3 Anatomical comparison between (A–C) the pelycosaur Secodontosaurus obtusidens (modified from Reisz, Berman & Scott, 1992) and (D–F) FC-DPV 2641 in (A, D) posterior, (B, E) left lateral, and (C, F) ventral views. Supraoccipitals, opisthotics and exoccipitals are indicated in blue, basioccipital in red, and parasphenoid in green.

Abbreviations: bo, basioccipital; bt, basal tubera; ds, damaged surface; eo, exoccipital; lf, lateral flange of the basioccipital; mwpr, medial wall of the pseudolagenar recess; np, notochordal pit; oc, occipital condyle; op, opisthotic; pp, paraoccipital process; ps, parasphenoid; so, supraoccipital; vrop, ventral ramus of the opisthotic. Scale bars equal 10 mm.

Figure 4 Schematic anatomical comparison between (A, B) FC-DPV 2641, (C) Protorosaurus speneri (modified from Gottmann-Quesada & Sander, 2009), and (D, E) a sub-adult specimen of Proterosuchus alexanderi (NMQR 1484) in occipital views.

Supraccopitals indicated in light brown, exoccipitals and opisthotics in blue, basioccipital in red, parabasisphenoid in green, pterygoid in white, and indeterminate bones or unossified areas in grey. Dashed areas in (B) indicate that they may represent basioccipital or exoccipitals but the condition cannot be determined because of co-osification. Abbreviations: XII?, possible exit of the hypoglossal cranial nerve; bo, basioccipital; bo?, possible basioccipital; bsp, basipterygoid process; bt, basal tubera; eo, exoccipital; eo?, possible exoccipital; fm, foramen magnum; fo, fenestra ovalis; lf, lateral flange of the basioccipital; mf, metotic foramen; mf?, possible metotic foramen; np, notochordal pit; oc, occipital condyle; p, parietal; pbbt, parabasisphenoid contribution to the basal tuebra; plr, pseudolagenar recess; pp, paraoccipital process; so, supraoccipital; vrop, ventral ramus of the opisthotic. Scale bars equal 10 mm.

Figure 5 Photographs and interpretive drawings of (A–C) an anterior cervical vertebra (FC-DPV 2640) and (D–H) a middle-posterior cervical vertebra (FC-DPV 2637) from the Late Permian–Early Triassic Buena Vista Formation (Uruguay) in (A–B, D–E) right lateral, (C, G) ventral, (F) dorsal, and (H) posterior views.

The arrows indicate the longitudinal ridge on the lateral surface of the centrum. Abbreviations: d, depression; dp, diapophysis; ns, neural spine; pa, parapophysis; pcdl, posterior centrodiapophyseal lamina; pdl, paradiapophyseal lamina; pfc, posterior facet of the centrum; podl, postzygodiapophyseal lamina; posf, postspinal fossa; prdl, prezygodiapophyseal lamina; prz, prezygapophysis; poz, postzygapophysis. Scale bars equal 5 mm.

Figure 6 Photographs and interpretive drawings of a middle-posterior cervical vertebra (FC-DPV 2639) from the Late Permian–Early Triassic Buena Vista Formation (Uruguay) in (A, B) anterior; (C, D) posterior; (E, F) right lateral; (G, H) left lateral; (I, J) dorsal; and (K, L) ventral views.

Abbreviations: afc, anterior facet of the centrum; ao, anterior overhanging; d, depression; dt, distal thickening; fr, facet for the rib; nc, neural canal; np, notochordal pit; ns, neural spine; pcdl, posterior centrodiapophyseal lamina; pfc, posterior facet of the centrum; posf, postspinal fossa; poz, postzygapophysis; prsf, prespinal fossa; prz, prezygapophysis. Scale bars equal 5 mm.

Description

Braincase. FC-DPV 2641 (Fig. 2; Table 1) is represented by an almost complete, slightly weathered basioccipital fused to the distal end of both exoccipitals. The presence of exoccipitals (Fig. 2: eo) is mainly inferred because the ventrolateral borders of the foramen magnum (which are preserved in FC-DPV 2641) are formed by these bones in other amniotans, such as basal synapsids (e.g., Romer & Price, 1940), parareptiles (e.g., Leptopleuron lacertinum Owen, 1851: (Spencer, 2000); Hypsognathus fenneri Gilmore, 1928: (Sues et al., 2000)) and archosauromorphs (e.g., Azendohsaurus madagaskarensis Flynn et al., 2010: UA 7-20-99-653; ‘Chasmatosaurus’ yuani Young, 1936: IVPP V2719; Doswellia kaltenbachi Weems, 1980: USNM 214823; Chanaresuchus bonapartei Romer, 1971: MCZ 4037). The fusion between the exoccipitals and basioccipital occurs through ontogeny in several groups of amniotans, including basal diapsids (e.g., Youngina capensis (Broom, 1914): TM 3603, Evans, 1987; Gephyrosaurus bridensis Evans, 1980; Mesosuchus browni (Watson, 1912): SAM-PK-6536, (Dilkes, 1998)) and basal synapsids (e.g., Secodontosaurus obtusidens (Cope, 1880): (Romer & Price, 1940; Reisz, Berman & Scott, 1992)) (Fig. 3). As a result, the presence of this condition in FC-DPV 2641 probably indicates that this specimen did not belong to, at least, an early juvenile.

Table 1 Measurements of the basioccipital + exoccipitals (FC-DPV 2641) in millimeters.

Values between brackets indicate incomplete measurements. Maximum deviation of the digital caliper is 0.02 mm but measurements were rounded to the nearest 0.1 millimeter.

Length of basioccpital	16.6	
Width of basioccipital	(17.1)	
Height of basioccipital	(13.6)	
Occipital condyle height	9.0	
Occipital condyle width	12.6	
Occipital condyle length	5.3	
Notochordal pit height	2.0	
Notochordal pit width	2.3	
Basal tuber length	8.5	
Basal tuber width	4.2	

The occipital condyle of FC-DPV 2641 is poorly posteriorly projected as a result of a short occipital neck (Fig. 2: oc), resembling the condition in several basal diapsids (e.g., Araeoscelis gracilis Vaughn, 1955; Gephyrosaurus bridensis Evans, 1980; Mesosuchus browni: SAM-PK-6536; Proterosuchus alexanderi (Hoffman, 1965): NMQR 1484; Prolacerta broomi Parrington, 1935: BP/1/2675; Archeopelta arborensis Desojo, Ezcurra & Schultz, 2011: CPEZ-239a), parareptiles (e.g., Hypsognathus fenneri (Sues et al., 2000)) and sphenacodont pelycosaurs (Romer & Price, 1940; Reisz, Berman & Scott, 1992). The occipital condyle is semi-spherical in overall shape, as occurs in archosauromorphs. Part of the posterior surface of the occipital condyle is flat, resembling the condition in the archosauromorphs Mesosuchus browni (SAM-PK-6536) and Prolacerta broomi (BP/1/2675), and some basal synapsids (e.g., Secodontosaurus obtusidens: Reisz, Berman & Scott, 1992) (Fig. 3). The occipital condyle has a shallow, sub-circular notochordal pit immediately below the ventral border of the foramen magnum (Figs. 2E: np, 4A and 4B). The shape and position of this pit closely resembles that of Youngina capensis (Gardner et al., 2010), Proterosuchus alexanderi (NMQR 1484), ‘Chasmatosaurus’ yuani (IVPP V2719) and some basal synapsids (e.g., Dimetrodon: (Romer & Price, 1940); Secodontosaurus obtusidens: (Reisz, Berman & Scott, 1992)) (Figs. 3 and 4: np). The articular surface of the occipital condyle is delimited laterally by an anteroposteriorly concave recessed surface that forms a slightly constricted occipital neck in ventral view (Figs. 2C and 2D: rs). This recessed surface is delimited anteriorly by a posteroventrally facing surface that belongs to the lateral flange of the basioccipital body (Figs. 2–4: lf). This lateral flange is well developed, resembling the condition in Prolacerta broomi (BP/1/2675), Proterosuchus spp. (BSPG 1934 VII 514; NMQR 880, 1484) and ‘Chasmatosaurus’ yuani (IVPP V2719), and may have overlapped at least partially the ventral ramus of the opisthotic in posterior view (Fig. 4E: lf). The occipital condyle is only differentiated from the ventral surface of the main body of the basioccipital by a gentle, transverse change in slope at the median line, resembling the condition in several amniotans (e.g., Youngina capensis: (Gardner et al., 2010); Prolacerta broomi: BP/1/2675; Proterosuchus alexanderi: NMQR 1484; Secodontosaurus obtusidens: (Reisz, Berman & Scott, 1992)).

The ventral surface of the basioccipital, immediately anterior to the occipital condyle, is slightly anteroposteriorly concave and lacks the median tuberosity present in Garjainia prima (Ochev, 1958; Gower & Sennikov, 1996). The basioccipital region of the basal tubera is almost completely preserved, but their ventral surfaces are weathered off (Figs. 2–4: bt). These structures are well developed and ventrally directed, resembling the condition in some basal archosauromorphs (e.g., Proterosuchus spp.: BSPG 1934 VII 514; NMQR 880, 1484; Fugusuchus hejiapanensis Cheng, 1980: (Gower & Sennikov, 1996)) and some basal synapsids (e.g., Dimetrodon: (Romer & Price, 1940); Haptodus garnettensis Currie, 1977: (Laurin, 1993)). By contrast, the main axis of the basioccipital portion of the basal tubera is usually lateroventrally directed in most archosauromorphs, such as Mesosuchus browni (SAM-PK-6536), Azendohsaurus madagaskarensis (UA 7-20-99-653), Prolacerta broomi (BP/1/2675), Sarmatosuchus otschevi Sennikov, 1994 (PIN 2865/68), ‘Chasmatosaurus’ yuani (IVPP V2719), Erythrosuchus africanus Broom, 1905a (NHMUK R3592), Euparkeria capensis Broom, 1913 (SAM-PK-5867), Archeopelta arborensis (CPEZ-239a) and Chanaresuchus bonapartei (PULR 07, MCZ 4037). The basal tubera are completely separated from each other at their bases, as also occurs in several diapsids (e.g., Youngina capensis: (Gardner et al., 2010); Gephyrosaurus bridensis: (Evans, 1980); Mesosuchus browni: SAM-PK-6536; Prolacerta broomi: BP/1/2675; Proterosuchus spp.: BSPG 1934 VII 514; NMQR 880, 1484; Euparkeria capensis: SAM-PK-5867; Chanaresuchus bonapartei: PULR 07, MCZ 4037). By contrast, in some other archosauromorphs the basal tubera are connected with each other by a transverse osseous lamina (e.g., Azendohsaurus madagaskarensis: UA 7-20-99-653; Trilophosaurus buettneri Case, 1928: (Spielmann et al., 2008); ‘Chasmatosaurus’ yuani: IVPP V2719; Fugusuchus hejiapanensis: (Gower & Sennikov, 1996); Erythrosuchus africanus: NHMUK R3592). In ventral view, the basal tubera are parallel to each other and to the sagittal plane of the basioccipital. The ventral surface of the basioccipital, between both basal tubera, is transversely concave and lacks the sub-circular foramen present in ‘Chasmatosaurus’yuani (IVPP V2719) and some specimens of Proterosuchus (NMQR 880).

The lateral surface of the basioccipital is subdivided into dorsolaterally and lateroventrally facing surfaces. Both surfaces meet each other in an obtuse angle in posterior view at the apex of the lateral flange of the bone. The dorsolaterally facing surface is flat and probably participated of the medial wall of the metotic foramen (Figs. 2C and 2E: wmf). The ventrolaterally facing surface is damaged on the left side of the bone (Fig. 3E: ds), but well preserved on the right side. The ventrolaterally facing surface has a complex topology and is subdivided by a diagonal, posteroventrally-to-anterodorsally oriented ridge (Fig. 2: dr). The facet for reception of the ventral ramus of the opisthotic is situated posterodorsally to this ridge (Fig. 2C: fvro). This facet is posterodorsally-to-anteroventrally slightly concave. The surface anteroventral to the diagonal ridge is more deeply anteroposteriorly concave than the facet for the ventral ramus of the opisthotic and is delimited anteriorly by the facet for the parabasisphenoid (Fig. 2: fpbs). The presence of smooth cortical bone on this deeply concave surface indicates that probably it was a non-articulating surface and potentially might have been part of the medial wall of the passage of the pseudolagenar recess (Fig. 2: mwpr). The pseudolagenar recess is present in Prolacerta broomi, Euparkeria capensis, several proterosuchian-grade archosauriforms and the poposauroid Xilousuchus sapingensis Wu, 1981 (Gower & Sennikov, 1996).

The anterior surface of the basioccipital has a slightly transversely convex facet for articulation with the parabasisphenoid (Fig. 2: fpbs). This articular facet extends also onto the anterodorsal surface of the basioccipital, immediately lateral to the floor of the endocranial cavity. The floor of the endocranial cavity is flat and has an anteroposteriorly long median longitudinal ridge, which is restricted to the anterior half of the basioccipital (Figs. 2B and 2E: mlr), resembling the condition in some procolophonids (e.g., Leptopleuron lacertinum (Spencer, 2000)), synapsids (e.g., Haptodus garnettensis: (Laurin, 1993)) and diapsids (e.g., Youngina capensis: (Gardner et al., 2010); Gephyrosaurus bridensis: (Evans, 1980)). The floor of the endocranial cavity of Prolacerta broomi has a pair of longitudinal ridges that delimit a shallow, median groove along most of the dorsal surface of the basioccipital (BP/1/2675).

The absence of suture between the right exoccipital and basioccipital precludes determining if the exoccipitals contact each other on the floor of the endocranial cavity. The foramen/foramina for the exit of the hypoglossal and glossopharyngeal cranial nerves (CN XI–XII) are not preserved.

Anterior cervical vertebra. FC-DPV 2640 (Figs. 5A–5C; Table 2) is interpreted as an anterior cervical vertebra because of its strong anteroposterior elongation and a facet for articulation with the rib (only the base of the left structure is preserved) placed next to the anterior margin of the neural arch. The anterior end of the centrum is damaged, but it seems to have been strongly bevelled and anteroventrally facing. If this condition is not an artefact due to damage, FC-DPV 2640 may correspond to an axis because it has enough room to receive the intercentrum of the axis. The posterior surface of the centrum is concave and seems to be slightly bevelled, possibly to receive a small intercentrum. The vertebra is possibly not notochordal. The centrum is approximately 3.6 times longer than tall, a ratio that closely resembles the condition in the axis of tanystropheids (e.g., Tanystropheus longobardicus Bassani, 1886: (Nosotti, 2007); Amotosaurus rotfeldensis Fraser & Rieppel, 2006) and the third and fourth cervical vertebrae of moderately long-necked basal archosauromorphs, such as Protorosaurus speneri (BSPG 1995 I 5, cast of WMSN P47361), Prolacerta broomi (BP/1/2675) and Macrocnemus bassanii (Nopcsa, 1930) (PIMUZ T2472, T4355, T4822). By contrast, the anterior cervical vertebrae of other basal archosauromorphs are proportionally shorter (e.g., Boreopricea funerea Tatarinov, 1978, PIN 3708/1: 1.92-2.00; Jesairosaurus lehmani (Jalil, 1997), ZAR 07: <2.00; Mesosuchus browni, SAM-PK-5882, fourth cervical: 2.01; Trilophosaurus buettneri, Spielmann et al. (2008: appendix 10): 1.84-2.50). The ventral surface of the centrum has a low and conspicuous median longitudinal keel. This keel extends along the entire preserved ventral surface of the centrum and becomes lower anteriorly. The centrum is slightly transversely compressed at mid-length and lacks a lateral fossa. The lateral surface of the centrum has a thin, longitudinal ridge that extends posteriorly from the base of the diapophysis to its posterior rim (Figs. 5A and 5B: arrow). A similar ridge is present in Macrocnemus bassanii (PIMUZ T4822), Tanystropheus longobardicus (PIMUZ T2818) and Eorasaurus olsoni (PIN 156/108, 109). A slightly developed longitudinal ridge is also present below the level of the diapophysis in some other basal diapsids (e.g., Protorosaurus speneri: (Gottmann-Quesada & Sander, 2009)). By contrast, the lateral surface of the centrum lacks a ridge in Petrolacosaurus kansensis Lane, 1945 (Reisz, 1981), Gephyrosaurus bridensis (Evans, 1981), Trilophosaurus buettneri (Spielmann et al., 2008), Prolacerta broomi (BP/1/2675) and Proterosuchus alexanderi (NMQR 1484). Only the base of the left facet for articulation with the rib is preserved and is restricted to the anteroventral portion of the centrum. The neurocentral suture is completely closed.

Table 2 Measurements of the anterior (FC-DPV 2640), middle-posterior (FC-DPV 2639), and middle-posterior (FC-DPV 2637) cervical vertebrae in millimeters.

Values between brackets indicate incomplete measurements and between squared brackets indicate estimated measurements. The length along the zygapophyses is the maximum anteroposterior length between the anterior tips of the prezygapophyses and the posterior tips of the postzygapophyses. Maximum deviation of the digital caliper is 0.02 mm but measurements were rounded to the nearest 0.1 millimeter.

	FC-DPV2640	FC-DPV 2639	FC-DPV 2637	
Centrum length	(17.8)	12.6	12.1	
Anterior facet of centrum width	-	4.4	[7.8]	
Anterior facet of centrum height	-	4.7	[7.1]	
Posterior facet of centrum width	4.8	(4.2)	(5.9)	
Posterior facet of centrum height	(4.9)	4.7	(6.5)	
Length along zygapophyses	(19.8)	(13.5)	(11.8)	
Height of neural spine	(3.1)	5.8	-	
Length of neural spine	(9.4)	8.9	4.7	
Maximum height of vertebra	(13.2)	14.0	(12.2)	

The zygapophyses lack their distal ends, but their preserved portions indicate that they were anteroposteriorly long, laterally divergent and sub-horizontal. As a result, the distal tips of the zygapophyses are well separated from the median line, resembling the condition in the anterior and middle cervicals of other basal archosauromorphs (e.g Trilophosaurus buettneri: (Gregory, 1945; Spielmann et al., 2008)). The neural arch has a very shallow depression lateral to the base of the neural spine (Figs. 5A and 5B: d), as occurs in Prolacerta broomi (BP/1/2675) and several other basal archosauromorphs. By contrast, the middle and posterior cervical vertebrae of at least some specimens of Proterosuchus alexanderi have a better defined and deeper, sub-circular fossa lateral to the base of the neural spine (NMQR 1484). The neural spine is mostly complete, but its dorsal margin is damaged where it becomes very thin transversely (Figs. 5A and 5B: ns). As a result, it is interpreted that the neural spine should not have been much taller and preserves it general shape. The neural spine is dorsoventrally short and strongly elongated anteroposteriorly, as occurs in Protorosaurus speneri (BSPG 1995 I 5, cast of WMSN P47361), Prolacerta broomi (BP/1/2675), Amotosaurus rotfeldensis (SMNS 50830) and Macrocnemus bassanii (PIMUZ T2472, T4355, T4822), but contrasting with the taller and anteroposteriorly shorter neural spine of Proterosuchus alexanderi (NMQR 1484).

Middle-posterior cervical vertebra. The degree of anteroposterior elongation and the presence of a parallelogram-shaped centrum indicate that FC-DPV 2639 (Fig. 6; Table 2) belongs to a middle or posterior cervical vertebra after comparisons with other basal archosauromorphs (e.g., Prolacerta broomi: BP/1/2675; Proterosuchus alexanderi: NMQR 1484; Trilophosaurus buettneri: (Spielmann et al., 2008)). The vertebra is moderately elongated anteroposteriorly, in which the length of the centrum is 2.68 times the height of its anterior articular surface. This ratio is slightly lower than that present in the middle cervical vertebrae of moderately long-necked basal archosauromorphs (>3.0, e.g., Prolacerta broomi: BP/1/2675; Macrocnemus bessanii: PIMUZ T4822; Protorosaurus speneri: BSPG 1995 I 5; Eorasaurus olsoni: PIN 156/108, 109). By contrast, the middle cervical vertebrae of Trilophosaurus buettneri (Spielmann et al., 2008), rhynchosaurs (e.g., Mesosuchus browni: SAM-PK-5882) and several basal archosauriforms (e.g., Proterosuchus alexanderi: NMQR 1484, Erythrosuchus africanus: NHMUK R3592; Euparkeria capensis: SAM-PK-586) are considerably proportionally anteroposteriorly shorter than FC-DPV 2639. The anterior articular facet of the centrum is more dorsally situated than the posterior one, resulting in a parallelogram-shaped centrum in lateral view (Figs. 6E–6H), as occurs in basal archosauromorphs (Ezcurra, Scheyer & Butler, 2014). The centrum is amphicoelous and apparently not notochordal (i.e., lacks a continuous canal piercing the centrum), contrasting with the condition present in basal synapsids, parareptiles, early diapsids, basal lepidosauromorphs and the basal archosauromorph Aenigmastropheus parringtoni (Ezcurra, Scheyer & Butler, 2014). The anterior articular facet is subcircular (Fig. 6: afc) and has a notochordal pit (Fig. 6: np). The posterior facet is damaged and its overall contour cannot be determined (Fig. 6: pfc), but the preserved portion is congruent in morphology with that of the anterior facet. The ventral surface of the centrum is strongly transversely convex along its entire extension and has a subtle median longitudinal edge (Figs. 6K and 6L). The centrum is incipiently transversely compressed at mid-length. The lateral surface of the centrum is continuously dorsoventrally convex and lacks a lateral fossa. The vertebra has a single, anteroposteriorly elongated facet for articulation with the rib (Fig. 6: fr), as occurs in non-archosauromorph diapsids and tanystropheids (e.g., Tanystropheus longobardicus: Wild, 1973). This facet is restricted to the anterior half of the vertebra and situated approximately at level with the centrum-neural arch boundary. The neurocentral suture is completely closed.

In the neural arch, a posterior centrodiapophyseal lamina delimits a central infradiapophyseal fossa below the base of the transverse process (Fig. 6: pcdl). In addition, a tuberosity runs from the base of the transverse process towards the base of the postzygapophysis, but it does not reach the latter structure. This tuberosity and the posterior centrodiapophyseal lamina delimit a shallow subtriangular depression that is topologically equivalent to a postzygapophyseal centrodiapophyseal fossa. There are no anterior centrodiapophyseal and prezygodiapophyseal laminae in the neural arch, which may be a result of the relatively anterior position of the vertebra in the cervical series. The zygapophyses are horizontal and anteroposteriorly short, but extend slightly beyond the margins of the anterior and posterior articular facets of the centrum, respectively. The prezygapophyses are anterolaterally directed and, as a result, their distal tips are well separated from the median line (Fig. 6: prz), as occurs in the cervico-dorsal vertebrae of Macrocnemus bessanii (PIMUZ T482), Tanystropheus longobardicus (Wild, 1973) and Trilophosaurus buettneri (Spielmann et al., 2008). The articular surfaces of the zygapophyses are damaged and it is not possible to determine their morphology. A shallow and poorly defined, circular depression is present laterally to the base of the neural spine (Fig. 6: d), as occurs in at least some specimens of Proterosuchus alexanderi (NMQR 1484). The neural spine is moderately low and strongly anteroposteriorly elongated, being considerably anteroposteriorly longer than tall (Fig. 6: ns), closely resembling the condition in Protorosaurus speneri (BSPG 1995 I 5), Macrocnemus bessanii (PIMUZ T4822) and Prolacerta broomi (BP/1/2675). By contrast, in Mesosuchus browni (SAM-PK-5882), Trilophosaurus buettneri (Spielmann et al., 2008), Proterosuchus fergusi Broom, 1903 (BSPG 1934-VIII-514; GHG 231), Sarmatosuchus otschevi (PIN 2865/13-19), Erythrosuchus africanus (NHMUK R3592), Garjainia prima (PIN 2394/5-13, 5-16) and Euparkeria capensis (SAM-PK-586) the neural spines are taller than long. The neural spine has an anterior overhang that extends anteriorly beyond the base of the spine (Fig. 6: ao), as occurs in Protorosaurus speneri (BSPG 1995 I 5), Macrocnemus bessanii (PIMUZ T4822), Trilophosaurus buettneri (Spielmann et al., 2008) and Prolacerta broomi (BP/1/2675). The distal margin of the neural spine has a low transverse thickening (Fig. 6: dt), but it does not form a spine table or a mammillary process. The same thickening on the distal margin of the neural spine is present in several other long-necked archosauromorphs (e.g., Macrocnemus bessanii: PIMUZ T4822; Prolacerta broomi: BP/1/2675). The pre- and postspinal fossae are deep and transversely wide (Fig. 6: posf, prsf). The prespinal fossa is restricted to the base of the neural spine and the postspinal fossa extends onto most of the posterior surface of the spine, as usually occurs in other basal archsoauromorphs (e.g., Prolacerta broomi: BP/1/2675).

Middle-posterior cervical vertebra. FC-DPV 2637 (Figs. 5D–5H; Table 2) belongs to a middle or posterior cervical vertebra because the parapophyses are situated slightly above the mid-height of the centrum, adjacent to its anterior margin (Fig. 5E: pa), and the neural spine is anteroposteriorly short (Figs. 5E and 5F: ns). This vertebra is well-preserved, but moderately squeezed posteroventrally to the right side (Fig. 5H), the posterior articular facet of the centrum is damaged and most of the prezygapophyses, right diapophysis and neural spine are missing. The centrum is amphicoelous and apparently not notochordal. The centrum length represents 1.7 times the height of its anterior articular facet, being proportionally shorter than FC-DPV 2639 and resembling the ratio present in the middle-posterior cervical vertebrae of several basal archosauromorphs (e.g., Aenigmastropheus parringtoni: UMZC T836; Eorasaurus olsoni: PIN 156/109; Trilophosaurus buettneri: (Spielmann et al., 2008); Proterosuchus alexanderi: NMQR 1484). The ventral surface of the centrum is transversely convex and lacks a median ventral keel. The centrum is slightly transversely compressed at mid-length. The anterior articular facet of the centrum is transversely broader than tall. The contour of the posterior facet cannot be determined because of damaging (Fig. 5H: pfc). The parapophyses are situated on laterally projected peduncles (Figs. 5D, 5E, 5G and 5H: pa). The peduncle of the parapophysis has a moderately deep depression on its ventral surface. The facet of the parapophysis is semi-circular, with a mostly straight anterior margin, and mainly laterally facing, with a low anteroventral component. A sub-horizontal ridge extends posteriorly from the base of the parapophysis to the lateral surface of the centrum, but it does not reach the level of mid-length of the centrum (Figs. 5D and 5E: arrow). A similar ridge is also present in FC-DPV 2640 (Figs. 5A and 5B) and the middle and posterior cervical vertebrae of other basal archosauromorphs, such as Macrocnemus bassanii (PIMUZ T4822), Tanystropheus longobardicus (PIMUZ T2818), Eorasaurus olsoni (PIN 156/108, 109) and Garjainia prima (PIN 2394/5-11, 5-13). The lateral surface of the centrum lacks a lateral fossa and the neurocentral suture is completely closed.

The diapophysis is mostly restricted to the anterior half of the neural arch (Fig. 5F: dp) and situated well above the centrum-neural arch boundary (Figs. 5D and 5E: dp). The diapophysis is moderately long and laterally developed, resembling the condition in other basal archosauromorphs (e.g., Prolacerta broomi: BP/1/2676). By contrast, in Eorasaurus olsoni and basal archosauriforms (e.g., Proterosuchus alexanderi: NMQR 1484) the diapophyses are better laterally developed than in FC-DPV 2637 (Ezcurra, Scheyer & Butler, 2014). The articular facet of the diapophysis is anteroposteriorly long, being considerably longer than tall. The neural arch has paradiapophyseal (Figs. 5D and 5E: pdl), posterior centrodiapophyseal (Figs. 5D and 5E: pcdl), prezygodiapophyseal (Figs. 5D–5F: prdl) and postzygodiapophyseal laminae (Figs. 5D, 5E and 5H: podl), as also occurs in the posterior cervical and anterior dorsal vertebrae of some basal archosauromorphs (e.g., Protorosaurus speneri: BSPG 1995 I 5; Tanystropheus longobardicus: PIMUZ T2817; Spinosuchus caseanus Von Huene, 1932: Spielmann et al., 2009) and several crown-archosaurs Butler, Barrett & Gower, 2012. By contrast, Prolacerta broomi has only anterior centrodiapophyseal/paradiapophyseal and prezygodiapophyseal laminae (BP/1/2675), and Proterosuchus spp. has anterior centrodiapophyseal/paradiapophyseal (NMQR 1484) and, in some specimens, postzygodiapophyseal laminae (SAM-PK-11208). The four laminae of FC-DPV 2637 delimit prezygapophyseal centrodiapophyseal, postzygapophyseal centrodiapophyseal, and centrodiapophyseal fossae. The zygapophyses are sub-horizontal and diverge slightly from the median line, resembling the condition in FC-DPV 2639 and FC-DPV 2640. The postzygapophysis (Figs. 5D–5F and 5H: poz) lacks epipophysis and its articular facet faces lateroventrally. There is a shallow fossa immediately lateral to the base of the neural spine (Fig. 5F: d), as occurs in FC-DPV 2639, Protorosaurus speneri and Proterosuchus alexanderi (NMQR 1484). The base of the neural spine is posteriorly displaced from the point of mid-length between the zygapophyses and subtriangular in cross-section, with an anteriorly oriented apex (Figs. 5D–5F: ns). The postspinal fossa is transversely broad and deep, and extends dorsally onto the entire preserved posterior surface of the neural spine (Fig. 5H: posf).

Taxonomic affinities

The partial braincase FC-DPV 2641 differs from those of parareptiles (e.g., Procolophon trigoniceps Owen, 1876: (Watson, 1914); Leptopleuron lacertinum: (Spencer, 2000); Owenetta kitchingorum Reisz & Scott, 2002; Hypsognathus fenneri (Sues et al., 2000)) in the combination of a proportionally anteroposteriorly long basioccipital body (anteroposterior length of the body (excluding occipital condyle and anterior projection between posterolateral processes of the basisphenoid) versus maximum transverse width = 0.64; whereas the same ratio is 0.29 in Leptopleuron lacertinum (Spencer, 2000), 0.40 in Owenetta kitchingorum (Reisz & Scott, 2002), and 0.23 in Hypsognathus fenneri (Sues et al., 2000)), transversely narrow exoccipitals, vertical basal tubera and a semi-spherical occipital condyle. In addition, FC-DPV 2641 differs from basal synapsids in the presence of anteroposteriorly long basal tubera, being considerably longer than broad, and a sub-spherical occipital condyle (Fig. 3). By contrast, the occipital condyle of most pelycosaurs has an extensive planar posterior surface, resulting in a sub-quadrangular structure in ventral or lateral view (Fig. 3B; but a sub-spherical occipital condyle is also present in Varanops brevirostris (Williston, 1911): Campione & Reisz, 2010), the basioccipital component of the basal tubera are strongly restricted posteriorly, being approximately as long as broad (Fig. 3C), and lacks an embayment to receive the massive footplate of the stape (e.g., Dimetrodon limbatus (Cope, 1877): (Romer & Price, 1940); Edaphosaurus pogonias Cope, 1882: (Romer & Price, 1940); Ophiacodon uniformis (Cope, 1878): (Romer & Price, 1940); Aerosaurus wellesi Langston & Reisz, 1981; Secodontosaurus obtusidens: (Reisz, Berman & Scott, 1992); Haptodus garnettensis: (Laurin, 1993); Varanops brevirostris: (Campione & Reisz, 2010)). Furthermore, the Uruguayan partial braincase differs from Permo-Triassic cynodont synapsids (e.g., Platycraniellus elegans (Van Hoepen, 1916): Abdala, 2007) in the presence of a single occipital condyle. Within Diapsida, FC-DPV 2641 differs from non-archosauromorph taxa (e.g., Araeoscelis gracilis: (Vaughn, 1955); Gephyrosaurus bridensis: (Evans, 1980); Planocephalosaurus robinsonae (Fraser, 1982); Youngina capensis: (Evans, 1987; Gardner et al., 2010)) in the presence of a semi-spherical occipital condyle and vertical basal tubera.

The presence of a semi-spherical occipital condyle and considerably anteroposteriorly longer than broad and vertical basal tubera is a combination of characters present only in Archosauromorpha, and allow the assignment of FC-DPV 2641 to this clade. No archosauromorph cranial remains have been described from other Permo-Triassic beds of South America (Dias-da-Silva, 1998; Da-Rosa et al., 2009; Dias-da-Silva & Da-Rosa, 2011), hampering comparisons with FC-DPV 2641. Similarly, we could not make comparisons with Late Permian archosauromorphs, such as Archosaurus rossicus and Protorosaurus speneri, because the braincase in these taxa is unknown or the knowledge of its anatomy is very limited (Sennikov, 1988; Gottmann-Quesada & Sander, 2009; Ezcurra, Scheyer & Butler, 2014). Indeed, in only one specimen assigned to Protorosaurus speneri the occipital region is exposed but it is badly preserved and does not allow making proper comparisons (Fig. 4C). In particular, among archosauromorphs, FC-DPV 2641 shares with the South African species of Proterosuchus (i.e., Proterosuchus fergusi: BSPG 1934 VIII 514; Proterosuchus alexanderi: NMQR 1484; Proterosuchus goweri Ezcurra & Butler, 2014: NMQR 880) and Fugusuchus hejiapanensis (Gower & Sennikov, 1996) the presence of vertical basal tubera (Figs. 4A, 4B, 4D and 4E). By contrast, in other basal archosauromorphs the basal tubera are lateroventrally oriented, being divergent from each other in posterior view (e.g., Azendohsaurus madagaskarensis: UA-7-20-99-653; Trilophosaurus buettneri: (Spielmann et al., 2008); Mesosuchus browni: SAM-PK-6536; Howesia browni Broom, 1905b: SAM-PK-5885; Prolacerta broomi: BP/1/2675; ‘Chasmatosaurus’ yuani: IVPP V2719; Sarmatosuchus otschevi: PIN 2865/68; Garjainia prima: PIN 951/60; Erythrosuchus africanus: NHMUK R3592). In addition, FC-DPV 2641 and the South African species of Proterosuchus differ from Fugusuchus hejiapanensis in the presence of basal tubera not connected to each other at their base. FC-DPV 2641 seems to differ from Proterosuchus (e.g., Proterosuchus alexanderi: NMQR 1484) in the presence of a broader contribution of the basioccipital to the floor of the endocranial cavity (Fig. 4). However, this possible difference should be taken with caution because of the strong degree of fusion between the exoccipitals and basioccipital in the Uruguayan specimen. In conclusion, FC-DPV 2641 resembles Proterosuchus in overall morphology and the presence of the vertical basal tubera is probably an apomorphy of a grade of basal archosauriforms, because Fugusuchus hejiapanensis has been recovered as a more crown-ward archosauriform than proterosuchids in a recent phylogenetic analysis (Ezcurra, Lecuona & Martinelli, 2010). Therefore, FC-DPV 2641 is interpreted as an indeterminate archosauromorph, possibly archosauriform (depending on the phylogenetic relationships of basal members of the clade that are currently in state of flux; Ezcurra, Butler & Gower, 2013), cf. Proterosuchidae.

The anterior cervical vertebra FC-DPV 2640 and middle-posterior cervical vertebra FC-DPV 2639 are assigned to an archosauromorph diapsid because of the following combination of characters: probable non-notochordal and anteroposteriorly elongated centra, a sub-horizontal ridge on the lateral surface of the centra, a shallow fossa immediately lateral to the base of the neural spines, and neural spines considerably anteroposteriorly longer than tall (Fig. 5). In addition, the centrum of FC-DPV 2639 is parallelogram-shaped in lateral view, a character that was found as a synapomorphy of Archosauromorpha (Ezcurra, Scheyer & Butler, 2014), and has an anterior overhang and a transversely thickened distal margin on the neural spine, features that occur together in the basal archosauromorphs Macrocnemus bessanii (PIMUZ T4822) and Prolacerta broomi (BP/1/2675). These anterior and middle-posterior cervical vertebrae differ from those of “pelycosaur” synapsids and araeoscelidian diapsids in the presence of a probable non-notochordal centrum and a lower and longer neural spine (Ezcurra, Scheyer & Butler, 2014). Among long-necked basal archosauromorphs, FC-DPV 2637 and FC-DPV 2640 differ from Prolacerta broomi and Trilophosaurus buettneri in the presence of a low longitudinal lateral crest that runs posteriorly from the base of the facet for articulation with the rib, from Macrocnemus bessanii in the absence of epipophyses (PIMUZ T4822), and from other tanystropheids in the presence of a proportionally anteroposteriorly shorter centrum (e.g. Amotosaurus rotfeldensis: SMNS 50830). Although the neural spines have damaged distal margins, they seem to have been dorsoventrally short. As a result, they may have differed from Protorosaurus speneri (BSPG 1995 I 5), which has tall neural spines. The combination of characters observed in FC-DPV 2640 is consistent with that present in basal archosauromorphs, such as Prolacerta broomi (BP/1/2675), but FC-DPV 2639 differs from this species in the presence of proportionally anteroposteriorly shorter centrum, and a less developed anterior overhang on the neural spine. FC-DPV 2639 and FC-DPV 2640 differ from the protorosaur cervical vertebra described by Dias-da-Silva (1998) from the Sanga do Cabral Formation in being considerably anteroposteriorly shorter. However, these differences could be due to the position of the vertebrae in the cervical series and the possibility that they belong to closely related species cannot be ruled out. Finally, the probable presence of a single facet for articulation with the cervical rib is a feature shared with non-archosauromorph diapsids (e.g., Gephyrosaurus bridensis: (Evans, 1981); Planocephalosaurus robinsonae: (Fraser & Walkden, 1984)) and tanystropheids (e.g., Amotosaurus rotfeldensis: SMNS 50830; Tanystropheus longobardicus: (Wild, 1973; Nosotti, 2007)). By contrast, more derived archosauromorphs have distinct parapophyses and diapophyses in the postaxial cervicals (e.g., Mesosuchus browni: (Dilkes, 1998); Trilophosaurus buettneri: (Spielmann et al., 2008); Prolacerta: BP/1/2675). Accordingly, FC-DPV 2639 and FC-DPV 2640 are interpreted as indeterminate basal archosauromorphs, but they might belong to a very basal member of the clade because of the presence of a single facet for the rib.

The middle-posterior cervical vertebra FC-DPV 2637 has a series of characters that were found as synapomorphies of Archosauromorpha or less inclusive clades within the group by Ezcurra, Scheyer & Butler (2014): a trapezoidal and probable non-notochordal centrum, and anterior centrodiapophyseal, posterior centrodiapophyseal, prezygodiapophyseal and postzygodiapophyseal laminae on the neural arch (Figs. 5D and 5E). The combination of characters observed in FC-DPV 2637 resembles that present in several disparate basal archosauromorphs (e.g., Protorosaurus speneri: BSPG 1995 I 5; Tanystropheus longobardicus: PIMUZ T2817; Spinosuchus caseanus: Spielmann et al., 2009). Nevertheless, FC-DPV 2637 differs from Prolacerta broomi and Proterosuchus fergusi in the presence of a posterior centrodiapophyseal lamina. FC-DPV 2637 cannot be properly compared with the protorosaur cervical vertebra from the Sanga do Cabral Formation (Dias-da-Silva, 1998) because the latter belongs to a more anterior element in the axial series. As a result, FC-DPV 2637 is interpreted as an indeterminate basal archosauromorph, but distinct from Prolacerta broomi and proterosuchids.

Discussion

The presence of archosauromorphs in the Permo-Triassic Buena Vista Formation was previously reported by Piñeiro (2002) and Piñeiro & Ubilla (2003), but no detailed description or taxonomic discussion have been provided so far. The cranial and postcranial remains described here increase the meagre archosauromorph record in Permo-Triassic rocks of South America.

The partial braincase with resemblances to Proterosuchus is particularly interesting because it probably reinforces the broad palaeobiogeographic distribution of proterosuchids during Permo-Triassic times (i.e., European Russia, China, South Africa and possibly Australia) (Ezcurra, Butler & Gower, 2013). However, we need to be cautious about this hypothesis because the specimen shows some differences that could not be properly compared with the Permian representatives of the group, such as the possible broad contribution of the basioccipital to the ventral margin of the foramen magnum (Fig. 4). The complete skull length of FC-DPV 2641 is estimated between 200–250 mm based on linear regressions of the total length of the skull versus the width and height of the occipital condyle, respectively, of a series of proterosuchid skulls from the Lystrosaurus Assemblage Zone of South Africa (N = 4: NMQR 880, 1484, BSPG 1934 VIII 514, GHG 231; occipital condyle width: y = 0.0514x–0.02115, R2 = 0.96; occipital condyle height: y = 0.0295x + 2.8892, R2 = 0.80). The skull length range recovered for FC-DPV 2641 falls in the 4th to 18th percentile of the South African proterosuchid sample (N = 14, total skull length ranges from 177 to 477 mm; Ezcurra & Butler, 2014). The presence of a medium-sized basal archosauromorph in the Permo-Triassic of South America is not unexpected because of the presence of relatively large proterosuchids in the latest Permian of Russia (Archosaurus rossicus) and relatively large basal archosauromorphs in the earliest Triassic of Antarctica (Smith et al., 2011), South Africa (Proterosuchus fergusi) and China (‘Chasmatosaurus’ yuani).

The archosauromorph partial braincase described here belongs to an individual considerably larger than those of the cervical vertebrae. Among the postcranial bones, FC-DPV 2637 pertained to an animal larger than that represented by FC-DPV 2639, 2640 and the latter two vertebrae are similar in size to each other (Table 2: compare transverse width of the centra). Thus, the archosauromorph bones described here should have belonged to multiple individuals. Furthermore, the simultaneous occurrence of a proterosuchid-like partial braincase and a middle-posterior cervical vertebra with clear differences with proterosuchids support the hypothesis of a multi-taxonomic archosauromorph assemblage. The archosauromorph record of the Buena Vista Formation seems to bolster a Permo-Triassic age for the unit, as previously concluded by Piñeiro, Ramos & Marsicano (2012). Unfortunately, this archosauromorph assemblage does not help in the debate of a Permian or Triassic age for this unit because both basal archosauromorphs and early archosauriforms (e.g., proterosuchids) are present across the Permo–Triassic boundary (Ezcurra, Scheyer & Butler, 2014).

We thank the following curators, researchers and collection managers that provided access to specimens under their care for the purpose of this research: Bernhard Zipfel, Bruce Rubidge and Fernando Abdala (BP); Markus Moser and Oliver Rauhut (BSPG); William Simpson (FMNH); Ellen de Kock (GHG); Liu Jun and Corwin Sullivan (IVPP); Jessica Cundiff (MCZ); Ronan Allain (MNHN); Sandra Chapman and Lorna Steel (NHMUK); Elize Butler and Jennifer Botha-Brink (NM); Heinz Furrer and Torsten Scheyer (PIMUZ); Andrey Sennikov (PIN); Sergio Martin (PULR); Sheena Kaal and Roger Smith (SAM); Rainer Schoch (SMNS); Heidi Fourie (TM); Mathew Lowe and Jennifer Clack (UMZC); and Michael Brett-Surman and Hans-Dieter Sues (USNM). We are grateful for the comments and suggestions of Sterling Nesbitt, María Jimena Trotteyn and Tiago Raugust, and the editor Mark Young.

Abbreviations

BP Evolutionary Studies Institute (formerly Bernard Price Institute for Palaeontological Research), University of the Witwatersrand, Johannesburg, South Africa;

BSPG Bayerische Staatssammlung für Paläontologie und Geologie, Munich, Germany

CPEZ Coleção Municipal, São Pedro do Sul; Brazil

FC-DPV Vertebrados Fósiles, Facultad de Ciencias, Montevideo, Uruguay

GHG Geological Survey, Pretoria, South Africa

IVPP Institute of Vertebrate Paleontology and Paleoanthropology, Beijing, China

MCZ Museum of Comparative Zoology, Cambridge, USA

NHMUK The Natural History Museum, London, UK

NM National Museum, Bloemfontein, South Africa

PIMUZ Paläontologisches Institut und Museum der Universität Zürich, Zurich, Switzerland

PIN Paleontological Institute of the Russian Academy of Sciences, Moscow, Russia

PULR Paleontología, Universidad Nacional de La Rioja, La Rioja, Argentina

SAM-PK Iziko South African Museum, Cape Town, South Africa

SMNS Staatliches Museum für Naturkunde Stuttgart, Stuttgart, Germany

TM Ditsong National Museum of Natural History (formerly Transvaal Museum), Pretoria, South Africa

UA University of Antananarivo, Antananarivo, Madagascar

UMZC University Museum of Zoology, Cambridge, UK

USNM National Museum of Natural History (formerly United States National Museum), Smithsonian Institution, Washington, D.C., USA

WMsN Westfälisches Museum für Naturkunde, Münster, Germany

ZAR Muséum national d’Histoire naturelle (Zarzaitine collection), Paris, France.

Additional Information and Declarations

Competing Interests

Author Contributions

Graciela Piñeiro is an Academic Editor for PeerJ.

Martín D. Ezcurra, Pablo Velozo, Melitta Meneghel and Graciela Piñeiro conceived and designed the experiments, performed the experiments, analyzed the data, contributed reagents/materials/analysis tools, wrote the paper, prepared figures and/or tables, reviewed drafts of the paper.

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
