# Peer review of "Early archosauromorph remains from the Permo-Triassic Buena Vista Formation of north-eastern Uruguay"

_PeerJ, doi:10.7717/peerj.776_

## Round 0.1 · original submission · Minor Revisions

The comments and annotated PDFs of the three reviewers give excellent advice on how to improve the manuscript.

I agree with the comments of reviewer one (María Trotteyn) that some additional figures and a preliminary phylogenetic would enhance the manuscript. See the comments of reviewer two (Sterling Nesbitt) on some potential phylogenetic characters.

I look forward to reading your revising manuscript.

·

Basic reporting

No Comments

Experimental design

No Comments

Validity of the findings

No Comments

Additional comments

I think that a preliminary phylogenetic analysis, geographical setting map and biostratigraphic column could enrich this manuscript.

·

Basic reporting

Overall, the paper is a good contribution and only requires some touch up. This is important information about the diversification of archosauromorphs across Pangea.

I only have a few comments about the figures. In general, I think that figures 2 and 3 need some refining.

Figure 2 - A-C have very high contrast and need to be lightened up. D-F should be replaced with the color figures in figure 1.

Figure 3 - Again, replace A with the same view from figure 1. The drawing in B could be a bit cleaner or is not needed at all. C is not needed because you cannot see much anyway. D is fine. E can be much better drawn - the lines could be a bit less wavy, the lines closer to the viewer should be thicker to give it some depth, and when two lines intersect, the line further away from the reader should be cut so it is clear which line is closer.

Experimental design

No comments

Validity of the findings

The description is solid. Is it possible that the anterior cervical vertebra is actually the axis? A ventral view of the specimen in figure 4 would be great. I suggest this because the anterior portion of the centrum looks like it is slightly damaged, but could be part of the facet for the atlas.

Taxonomic affinities of the braincase. Can any of your observations about the similarities of the new specimen and archosauromorphs be turned into phylogenetic characters? I think these characters could be useful, but without a more formal way to examine them, they are just similarities.

Exoccipitals meeting at the midline of the basioccipital - do they meet or do they not? It is hard to tell based on the comparative description and the taxonomic affinities section. This is an important character in Diapsida and can help with the archosauromorph placement.

Additional comments

line 28, replace "on" with "of"
line 28, what does "dominance" mean? Diversity, abundance?
line 29, "null" or "very meagre" should be replaced with poor or essentially nonexistent
line 87, remove hard return
line 118, "well ossified occipital condyle" is not the descriptor here. I think you mean coossified exocciptial and basioccipital.
line 121, replace "reduced" with "short". The use of reduced here implies it was longer previously (or outside the taxon).
line 202, Is the suture there or not?
line 231-233, It is very difficult to estimate the age of an animal based on a single neurocentral fused vertebra because you do not know the fusion pattern (head to tail or opposite). Can you use the sutural fusion pattern in a proterosuchid? If not, I don't think you can use this even for a rough estimate.
line 238, add Gregory 1945 here please.
line 397-398, "null" should be replaced with "unknown"
line 501, would you expect to see autapomorphies in an axial series that are carried through the column? I can't think of any examples off hand.

·

Basic reporting

No Comments

Experimental design

No Comments

Validity of the findings

No Comments

Additional comments

Congratulations! The discovery of this assemblage is an important scientific contribution since the knowledge of the Permo-Triassic archosauromorph record is currently null or poorly known in several continents (including South America). This work deserves and should be published after the revisions in the PDF document attached.

---

## Round 0.2 · Minor Revisions

Dear authors,

I very much like the amended version of your manuscript. After reading it, I have only a few minor suggested changes. Namely:
1. re: reviewer one, I think it best to use the nominal authority after the first use of a binomen (it is not necessary to use the nominal authority after every instance). While this may no longer be a requirement in most journals, as you pointed out, the comment is correct. Can the necessary changes be made.
2. 'northeastern'. Can you change that in the title and text to: 'north-eastern'. There are a few other compound nouns that could be hyphenated, such as 'crownward'.
3. 'Possess'. I understand what you're meaning (I myself have used possess in the past). But in anatomical descriptions, possess isn't really what is meant (i.e., a morphology isn't obtained/owned by the bone in question). Try to use another word, or rewrite the sentence to avoid using 'possess'.
4. I understand that the geographic map and biostratigraphic chart have already been published numerous times, but as the reviewers state, having in your manuscript would enrich it for readers. As PeerJ has no limit on page length or figure number, there is no reason not use that to help elucidate your work as much as possible.

After you have made these minor changes, I cannot forsee any reason why your manuscript should not be acceptable for publication in PeerJ.

Yours sincerely,
Mark Young

---

## Round 0.3 · accepted · Accept

Dear authors,

Thank you for the swift revision of your manuscript, and I am delighted to accept it for publication at PeerJ. I hope you consider publishing with PeerJ in the future.